# Fluorescent Carbon Dots from Food Industry By-Products for Cell Imaging

**DOI:** 10.3390/jfb14020090

**Published:** 2023-02-07

**Authors:** Federica Mancini, Arianna Menichetti, Lorenzo Degli Esposti, Monica Montesi, Silvia Panseri, Giada Bassi, Marco Montalti, Laura Lazzarini, Alessio Adamiano, Michele Iafisco

**Affiliations:** 1Institute of Science, Technology and Sustainability for Ceramics (ISSMC), National Research Council (CNR), 48018 Faenza, Italy; 2Department of Chemistry “Giacomo Ciamician”, Alma Mater Studiorum-Università di Bologna, 40126 Bologna, Italy; 3Department of Neuroscience, Imaging and Clinical Sciences, University of Studies “G. D’Annunzio”, 66100 Chieti, Italy; 4Institute of Materials for Electronics and Magnetism (IMEM), National Research Council (CNR), 43124 Parma, Italy

**Keywords:** carbon dots, food by-products, circular economy, photoluminescence, bioimaging, pH sensing

## Abstract

Herein, following a circular economy approach, we present the synthesis of luminescent carbon dots via the thermal treatment of chestnut and peanut shells, which are abundant carbon-rich food industry by-products. As-synthesized carbon dots have excellent water dispersibility thanks to their negative surface groups, good luminescence, and photo-stability. The excitation–emission behaviour as well as the surface functionalization of these carbon dots can be tuned by changing the carbon source (chestnuts or peanuts) and the dispersing medium (water or ammonium hydroxide solution). Preliminary in vitro biological data proved that the samples are not cytotoxic to fibroblasts and can act as luminescent probes for cellular imaging. In addition, these carbon dots have a pH-dependent luminescence and may, therefore, serve as cellular pH sensors. This work paves the way towards the development of more sustainable carbon dot production for biomedical applications.

## 1. Introduction

Circular economy is a production and consumption model that aims to reuse and recycle existing materials and products as far as possible [1]. This model was proposed around the early 2000s in contrast to the standard linear economy concept, which is based on the linear flow of materials from resources to products and then directly into waste [2]. Thus, circular economy aims to minimize waste through reusing, refurbishing, and recycling both raw materials and products. In addition, it aims to replace primary materials—such as fossil carbon sources—with secondary ones of suitable quality—for instance biomass wastes—thus economising on non-renewable resources [3].

Biomass waste includes a variety of materials, such as agricultural crops and residues, forestry wastes, animal processing wastes, and industrial and food residuals. Food by-products—i.e., cereal residues, nutshells, vegetables, and fruit peels—are an important carbon source to recover, as they contain a high fraction of organic/carbonaceous material [4]. In particular, nutshells are an abundant carbon-rich food industry by-product that is usually discarded, thus becoming unrecycled waste.

Nutshells are a subclass of lignocellulose biomass that mainly contain cellulose (25–30%), hemicellulose (22–28%), and lignin (30–40%), and are, therefore, mostly composed of carbon and oxygen [5,6]. For this reason, nutshells have recently emerged as an alternative and promising feedstock for the production of biofuels and biopolymers, as well as pesticides and fertilizers [7]. In addition to these applications, nutshells are also promising raw carbon sources for the production of photoluminescent materials [4,8].

Among several types of carbon-based nanomaterials—such as graphene, carbon nanotubes and nanofibers—fluorescent carbon dots (C-dots) have raised particular interest in the scientific community for their tuneable photophysical properties, as well as for their ease of fabrication and low cost [9]. C-dots are quasi-spherical nanoparticles usually less than 10 nm in size that possess strong luminescence properties [4]. C-dots display photophysical characteristics similar to traditional inorganic quantum dots (TQDs), such as good luminescence and tuneable emission. However, they possess unique advantages over TQDs such as improved water dispersibility, lower toxicity, and better biocompatibility and biodegradability [10,11,12,13]. For these reasons, there is a high interest in C-dots, especially in the biomedical field, to be used as drug delivery vehicles, as well as for cell imaging and in vivo bioimaging [14,15]. Moreover, C-dots can be employed for advanced biomedical applications such as hyperthermia and pH sensing, as they often possess temperature- or pH-dependent photoluminescence [16,17].

C-dot preparation can be classified into two main approaches: (i) top-down and (ii) bottom-up methods [9]. Top-down methods involve the conversion of bulk carbon materials (e.g., graphene, graphite, carbon nanotubes) into carbon nanoparticles, and include laser ablation, electrochemical oxidation, and arc discharge. The obtained C-dots usually possess a graphite-like structure and are characterized by low luminescence quantum yields. Top-down processes generally require harsh conditions and complex procedures, which strongly limits their application. Therefore, the most promising approaches to produce C-dots are the bottom-up methods, where C-dots are obtained through the carbonization of small molecules such as citric acid or dopamine [18,19]. Some of the most relevant bottom-up techniques for C-dot production are microwave-assisted synthesis, combustion, pyrolysis, solvothermal, and ultrasonic treatment. C-dots obtained through bottom-up approaches usually have an amorphous structure and high quantum yields. Moreover, the synthetic processes are often easy to perform and cost-effective.

Biowastes can also be used as carbon sources for the preparation of sustainable C-dots, following a bottom-up approach [20,21]. In the last few years, several food wastes were investigated to produce C-dots via extraction of the organic components and subsequent conversion into C-dots through several purification steps. In this regard, nutshell wastes are a promising resource for the production of C-dots due to their high carbon content and abundance.

In this work, we produced C-dots for bioimaging through a combustion reaction of nut by-products. In particular, we report the synthesis and characterization of blue-emitting C-dots obtained from two different nut sources—chestnut and peanut shells—through a simple and reliable carbonization and extraction process. Compared to other previously reported works [20,22], we investigated in detail the effect of two different carbon sources as well as that of two aqueous-based solvents on the chemical and photophysical properties of C-dots. The main biomedical applications of the produced C-dots as well as future perspectives are discussed. Furthermore, the whole C-dot production process was designed following a green approach, as biowastes are the sole carbon source and no organic solvents are required during the synthesis.

## 2. Materials and Methods

### 2.1. Materials

Chestnuts (C) and peanuts (P) were purchased from a local market and the shells were separated manually. Ultrapure water was produced with an Arium© Pro instrument from Sartorius (Goettingen, Germany). Ammonia aqueous solution (NH_4_OH, ACS reagent, 28–30% solution) was purchased from Merck. 9,10-Diphenylanthracene (>99%) was purchased from Ega-chemie.

### 2.2. Synthesis of C-Dots

The synthesis of fluorescent C-dots involves two steps: (i) the carbonization of C and P shells, and (ii) the purification of the dispersion.

In the first step, C and P shells were ground at 5000 rpm for 10 min using a blade mill (IKA MultiDrive basic, IKA^®^-Werke GmbH, Staufen, Germany) equipped with blades for fibrous materials (IKA MultiDrive accessories: MultiDrive MI 250 Milling chamber and MultiDrive MI 250.2 star-shaped cutter), and then sieved down to 600 µm. Subsequently, the sieved powder was thermally treated at 300 °C for 2 h under air atmosphere (300 °C, 2.5 °C/min heating ramp, 2 h dwell time) (Nannetti—model KL 20/13, Faenza, Italy). The obtained product was dispersed either in ultrapure water or in 0.1 M NH_4_OH and sonicated using a tip sonicator for 30 min, 40% amplitude, 5 s of pulse and 1 s of pause (VCX 500, SONICS & MATERIALS INC., Newtown CT, USA). Afterwards, the suspension was centrifuged (14,000 rpm for 10 min) and the supernatant containing C-dots was dialyzed against ultrapure water for 48 h (membrane MWCO = 3500 Da). After that, a further centrifugation step was performed (14,000 rpm for 15 min) and the resulting dispersion was filtrated over a 0.45 µm syringe filter. An aliquot of the purified dispersion was collected and freeze-dried for structural and morphological characterization. Samples were named on the basis of the nut product (C or P) and of the dispersing medium (H_2_O or NH_4_OH) as follows: (i) C H_2_O and P H_2_O, and (ii) C NH_4_OH and P NH_4_OH, for C-dots obtained from C and P shells using H_2_O or NH_4_OH solution as the extraction solvent, respectively.

### 2.3. C-Dots Characterization

Freeze-dried C-dots were characterized using powder X-ray diffraction (PXRD), Fourier transform infrared spectroscopy (FT-IR), and transmission electron microscopy (TEM). The PXRD pattern was collected with a D8 Advance diffractometer (Bruker, Karlsruhe, Germany). Cu Kα X-rays were generated at 40 kV and 40 mA. The pattern was collected in the 10–60° 2θ range with a step size of 0.02° 2θ and a counting time of 0.5 s. FT-IR spectra were collected in attenuated total reflectance (ATR) mode with a Nicolet iS5 spectrometer (Thermo Fisher Scientific Inc., Waltham, MA, USA) using an iD7 diamond ATR accessory. The spectra were collected with a resolution of 4 cm^−1^ by accumulation of 32 scans covering the 4000 to 400 cm^−1^ spectral range. The TEM micrographs were collected using a field-emission JEOL 2200FS electron microscope (JEOL Ltd., Akashima, Tokyo, Japan) operating at 200 kV and equipped with energy-dispersive spectroscopy (EDS). The samples were prepared by drop casting or spraying of the powder suspension in propanol onto ultra-thin carbon-coated copper grids.

C-dot aqueous dispersions were analysed through dynamic light scattering (DLS) to determine the hydrodynamic diameter distribution and electrophoretic mobility (ζ-potential). DLS and ζ-potential analyses were performed using a Zetasizer Nano ZSP Instrument (Malvern Instruments Ltd., Malvern, UK). The hydrodynamic diameter distribution of the samples was measured using the carbon dot refractive index (1.55) and water refractive index (1.33) as working parameters for the samples and the solvent, respectively. Size results are reported as the Z-average of the hydrodynamic diameter of the particles of three measurements at 25 °C of at least 10 runs. ζ-potentials were quantified as the electrophoretic mobility at 25 °C of three separate measurements (maximum 100 runs each) using laser Doppler velocimetry using a disposable electrophoretic cell (DTS1061, Malvern Ltd., Worcestershire, UK) with the same sample and solvent parameters.

Photophysical characterization of C-dots was performed both on as-synthesized suspensions as well as on freeze-dried samples redispersed in double-distilled water. C-dot absorption spectra were recorded using a Perkin Elmer Lambda 650 UV/Vis spectrometer (Perkin Elmer, Shelton, CT, USA). Emission spectra and 3D excitation and emission maps were recorded using a Horiba Fluoromax-4 spectrofluorometer (Horiba, Edison, NJ, USA).

Emission lifetimes were obtained through time-correlated single photon counting (Edinburgh Instruments, Levington, UK). Fluorescence quantum yields were calculated using 9,10-diphenylanthracene as the standard, by means of the following equation [23]:(1)$$\Phi =\Phi_{s}\frac{I}{I_{s}}\frac{A_{s}}{A}\frac{n^{2}}{n_{s}^{2}}$$
where *Φ* is the quantum yield, *Φ_s_* is the quantum yield of the standard, *I* is the integrated area of the emission spectrum of the sample and *I_s_* is that of the standard, *A* is the absorbance at the excitation wavelength of the sample and *A_s_* is that of the standard, and *n* is the refractive index of the solvent of the sample and *n_s_* is that of the standard. Absorption and emission spectra, 3D excitation/emission maps, emission lifetimes, and emission quantum yields were obtained for C-dot dispersions at 5 ppm concentration.

### 2.4. Biological Evaluation

#### 2.4.1. Cell Culture

Murine fibroblasts BALB/3T3 clone A31 cell line purchased from American Type Culture Collection (ATCC^®^ CCL-163^TM^) were used. Cells were cultured in Dulbecco’s modified Eagle medium (DMEM) with high glucose and pyruvate (Gibco) supplemented with 10% calf bovine serum (Gibco) and 1% penicillin/streptomycin mixture (pen/strep) (100 U/mL–100 µg/mL, Gibco). Cells were kept in an incubator at 37 °C under controlled humidity and 5% CO_2_ atmosphere conditions. Cells were detached from culture flasks by trypsinization and centrifuged. The cell number and viability were determined using the trypan blue dye exclusion test and all cell handling procedures were performed under a laminar flow hood in sterility conditions. For the experiment, the cells were seeded at a density of 15.625 cells/cm^2^ in 96-well plates.

#### 2.4.2. C-Dots In Vitro Testing

C-dots were added to cell cultures at the concentrations of 0.05, 0.125, 0.25, and 0.5 µg/µL in culture media starting from a stock 2.5 µg/µL dispersion of freeze-dried particles in 67 mM sodium citrate aqueous solution. The control group was cells only. The biological interaction between cells and C-dots was evaluated in terms of cell viability at 24 h of incubation. The cellular uptake of C-dots was analysed at 6 h of incubation.

#### 2.4.3. Cell Viability

A quantitative analysis of cell viability was carried out using an MTT assay on cell cultures, by using the cells only as the control group. At 24 h, the MTT assay was performed according to manufacturer’s instructions. Briefly, MTT reagent [3-(4,5-dimethylthiazol-2-yl)-2,5-diphenyltetrazolium bromide] (5 mg/mL) was dissolved in phosphate saline buffer 1X (PBS 1X). At 72 h, the cells were incubated with 10% MTT solution for 2 h at 37 °C, 5% CO_2_, and controlled humidity conditions. The cell culture medium was removed and substituted with 200 µL DMSO (Sigma, St. Louis, MI, USA), dissolving formazan crystals derived from MTT conversion by metabolically active cells. After 15 min incubation under slight stirring conditions, the absorbance of formazan was red at 570 nm by using a Multiskan FC Microplate Photometer (Thermo Scientific, Waltham, MA, USA). The values of absorbance are directly proportional to the number of metabolic active cells in each well. One experiment was carried out and a biological triplicate for each condition was performed, and cell growth under standard conditions without C-dots, named cells only, was used as the control group.

#### 2.4.4. Cellular Uptake

The cellular uptake of C-dots was qualitatively detected at 6 h of incubation after PBS 1X washing to remove the excess material by using a Ti-E Inverted Fluorescent Microscope (Nikon, Tokyo, Japan) with the DAPI filter. One experiment was carried out and a biological duplicate for each condition was performed at both time points.

#### 2.4.5. Statistical Analysis

The statistical analysis was performed by using GraphPAD Prism software (version 8.0.1). The cell viability data were reported in the graph as percentage mean (%) with respect to cells only ± standard error of the mean, and they were analysed with two-way analysis of variance (two-way ANOVA) and Tukey’s multiple comparisons test.

## 3. Results and Discussion

### 3.1. Preparation and Physico-Chemical Characterization

C and P shells are mainly composed of lignocellulose, meaning they are primarily composed of carbon and oxygen [24,25] and, therefore, can be exploited to produce C-dots. Among the plethora of bottom-up methods to prepare C-dots, carbonization under controlled conditions was chosen, as it is a simple and cost-effective process that does not require the use of organic solvents. Treatment temperature is the most crucial parameter in the bottom-up synthesis of C-dots from biowastes, as it influences their optical properties [9]. Indeed, an extremely high temperature would result in the over-carbonization of the surface of C-dots, leading to poor luminescence properties. On the other hand, a low reaction temperature would instead cause insufficient carbonization of the carbon nanoparticle core and, thus, lead to C-dots with a weak fluorescence emission [9,11]. Therefore, 300 °C was chosen as treatment temperature to obtain a well-carbonized material without affecting its surface properties.

After calcination, the product was dispersed either in H_2_O or NH_4_OH solution, followed by sonication and purification by means of dialysis and centrifugation (Figure 1). NH_4_OH was used as an alkaline nitrogen source to check whether the dispersion solvent affects the photo-physical properties of C-dots.

The Z-average of the particles determined through DLS is reported in Table 1. The hydrodynamic diameter of C-dots ranges between 100 and 170 nm, with the two samples dispersed in NH_4_OH solution having a higher average diameter than the ones in water. Such values of the hydrodynamic diameter are higher than those usually reported in the literature [9] and may be explained by taking into account the fact that we studied as-synthesized C-dots without performing any post-synthesis surface functionalization to improve their colloidal stability. Therefore, naked C-dots can easily aggregate to form clusters in the aqueous-based solvent, as previously reported in the literature [26]. This tendency towards aggregation was also confirmed through TEM analysis, as can be clearly observed in Appendix A. Nevertheless, infrequent smaller carbon nanoparticles between 10 and 20 nm in size can also be found, as evidenced by selected TEM images and relative EDS maps (Appendix A).

The high Z-average values of the four samples could also be explained by considering the pH of the individual dispersions. Indeed, it has been reported in the literature that pH values lower than 6 can induce C-dot aggregation into clusters and, therefore, account for the high Z-average values [27,28,29]. Nevertheless, the polydispersity (PdI) values are <0.4 and similar for all C-dot suspensions, indicating a narrow particle size distribution for all materials. Hence, future works will be focused on the introduction of functional surface groups onto C-dots to obtain a long-term stable colloidal system [30].

The ζ-potentials of C-dots are also reported in Table 1. All samples have negative ζ-potential values ranging between −20 and −50 mV, which indicates the presence of negatively charged functional groups on the C-dots’ surface. C-dots obtained from C have a more negative surface charge compared to the ones derived from P, suggesting a different nature or abundance of surface charged functional groups depending on the source material.

The FT-IR spectra of the C-dot samples are shown in Figure 2. The C-dots prepared in H_2_O have similar spectra and show four main peaks at 3210, 1557, 1378, and 1117 cm^−1^ [31]. In particular, the presence of the carboxyl group was confirmed by the characteristic absorption band of O–H at 3210 cm^−1^ and the stretching vibration band of C–O at 1117 cm^−1^ [32]. The sharp peak at 1557 cm^−1^ can be ascribed to C=O bond stretching, thus indicating the existence of carbonyl groups. Furthermore, the peak at 1378 cm^−1^ is associated with the bending vibrations of C=C, suggesting the presence of aryl groups [33]. The FT-IR spectra of the C-dots prepared in NH_4_OH show two additional peaks at 1250 and 768 cm^−1^, in comparison to the ones in H_2_O. The peak at 1250 cm^−1^ can be attributed to the C-O stretching of an epoxy/ether group, while the one at 768 cm^−1^ could be ascribed either to C=C or C-H bending, suggesting that in the presence of ammonia, C-dots have a more graphitic composition. Moreover, the absorption band at 1117 cm^−1^ is weakened for the C-dots prepared in NH_4_OH, especially in the case of C NH_4_OH, thus indicating a lesser abundance of carboxyl moieties in these samples.

Therefore, FT-IR data reveal that a different surface functionalization of C-dots can be achieved by varying the dispersing medium, where water induces a major abundance of carboxyl moieties, while ammonia generates a more graphite-like composition. In addition, the presence of carboxyl groups evidenced by FT-IR strengthens the ζ-potential results, demonstrating that C-dots’ surface contains negatively charged groups, which strongly contributes to their water dispersibility.

The PXRD pattern of C-dots is reported in Figure 3. For all the samples, only a broad band at 30° 2θ is observed, which is characteristic of an amorphous carbon phase [21,22]. This result is also in agreement with the TEM analysis. Moreover, this outcome is in line with previous results, as it is widely reported in the literature that C-dots synthesized from both natural and synthetic carbon sources commonly have an amorphous structure [22,31,34,35]. In the case of the PXRD data, there are no marked differences between the samples, neither on the basis of the source material nor of the dispersion solvent.

### 3.2. Photophysical Characterization

The optical properties of the C-dots were investigated using UV–Vis spectroscopy and photoluminescence studies. Overall, C-dot aqueous suspensions are yellowish under daylight irradiation (Figure 4A), and they emit blue light under 365 nm UV lamp irradiation (Figure 4B).

#### 3.2.1. Absorption Spectra

C-dots typically possess one or more strong ultraviolet absorption bands between 220–270 nm and 280–350 nm, but the positions of their UV absorption peaks vary considerably between the ultraviolet and the visible wavelength region, according to both the production process and the carbon-based precursor [9,36].

In our case, the absorption spectra of the four samples have the typical absorption profile of C-dots: a strong peak in the UV region around 200 nm and a band extending to the visible range [11,37]. In particular, the shoulder around 250 nm can be attributed to the π-π* transition of C=C bonds [35,37,38], while the tail in the visible range is attributed to the transitions of the surface states that usually depend on surface functionalities (Figure 4B) [39].

When comparing C and P C-dot dispersions (Figure 5A) and those after freeze-drying and subsequent redispersion in water (Figure 5B), the absorption spectra reveal a difference in concentration between the native dispersions and the freeze-dried samples. Indeed, although the nominal concentration is the same (5 ppm), the absorbance of the freeze-dried samples is generally lower than that of the native dispersions, except in the case of C H_2_O.

C H_2_O samples—especially the freeze-dried one—present a lower contribution of the absorption band in the visible range than the other C-dots, as shown in the normalized absorption spectra (insets in Figure 5A,B). This feature could be related to different surface functional groups of the C H_2_O samples in comparison to the other ones.

Moreover, the difference in size between C H_2_O/P H_2_O and C NH_4_OH/P NH_4_OH could also explain their spectral differences because of quantum confinement effects [40,41]. According to the quantum confinement effect, the absorption spectrum is shifted to lower wavelengths as the particle becomes smaller, which is in agreement with the particle size results. However, it must be mentioned that the correlation between C-dots’ photophysical properties and the quantum confinement effect has not always been confirmed in the literature, and, thus, the optical properties of C-dots might be mainly related to their surface states [42].

#### 3.2.2. Excitation–Emission Maps

Excitation–emission maps of C-dot native dispersions are reported in Figure 6. All samples absorb near-UV light (300–380 nm) and have a blue emission between 400 and 500 nm. The samples present the characteristic emission properties of C-dots: longer excitation wavelengths lead to red-shifted emission spectra [10]. This behaviour can be easily observed in the reported excitation–emission maps, where the emission area is tilted towards the top-right corner. The same emission properties are observed for the resuspended freeze-dried C-dots (Appendix A).

The normalized excitation–emission maps reveal that the C and P C-dot samples have different photophysical behaviours. While P H_2_O and P NH_4_OH have a markedly red-shifted excitation-dependent emission (λ_exc_/λ_em_: 340 nm/450 nm), this feature is less pronounced for the C NH_4_OH and C H_2_O samples, whose excitation and emission maxima are at lower wavelengths (λ_exc_/λ_em_: 320 nm/420 nm). Furthermore, a difference in the shape of the maps is observed, comparing the samples dispersed in H_2_O with the ones in NH_4_OH. The excitation–emission maps, therefore, indicate that both the starting material (chestnut or peanut) and the way in which the material is treated (H_2_O or NH_4_OH as dispersing medium) influence the excitation-wavelength-dependent emission of C-dots and can, therefore, be used to tune their properties. It has been widely reported in the literature that the photoluminescent properties of C-dots depend on their surface emissive states [28,43]; therefore, both the starting material and the dispersion solvent must have influenced the nature of such emissive sites.

#### 3.2.3. Quantum Yield and Emission Lifetime

Quantum yields were calculated at 450 nm to have a comparison of the emission properties of the different C-dots, and they are reported in Table 2. In particular, QYs were obtained maintaining the same concentration for all samples (5 ppm), which is also the same as the one used for 3D excitation/emission maps; in this way, the possible concentration effects on their photophysical properties are avoided [44]. C H_2_O and P H_2_O have higher emission quantum yields with respect to the NH_4_OH-treated samples. This suggests that the modification of C-dot surface functionalities using ammonia as the dispersing medium decreases the luminescence, likely due to the activation of non-radiative transitions. On the other hand, there is no clear correlation between the QY and the nature of the carbon source. However, it should be noted that the QY increases after freeze-drying and redispersion for H_2_O samples, which deserves to be investigated in more detail in the future. Overall, QYs are in the range 1–4%, so they are slightly lower than the QYs of other C-dots prepared with a similar approach, likely due to the different carbonization conditions [22].

Emission lifetime measurements showed that two different emission bands were present at 450 nm and 520 nm, suggesting the existence of multiple emissive states. Emission lifetimes at both wavelengths are listed in Table 2. For all samples, the lifetime at both wavelengths is about 4 ns, except for C H_2_O, where a higher lifetime is observed at 520 nm (5 ns). These results are indicative of fluorescence phenomena and are in agreement with the lifetimes of other C-dots reported in the literature [20]. Finally, there is almost no difference between the lifetime of the native dispersions and that of freeze-dried samples.

#### 3.2.4. pH-Dependent Excitation–Emission Maps

C-dots have a complex structure, with graphite or amorphous carbon as the main skeleton and the surface exposing carboxyl, hydroxyl, carbonyl, amino, and other functional groups that may undergo protonation and deprotonation. Therefore, the photoluminescence of C-dots may be pH-dependent, although little is known about the mechanisms regulating their pH sensitivity [38].

In this regard, we have discovered that our C-dots possess pH-sensitive luminescence, as can be clearly observed in Appendix A. As the pH increases from 3 to 11, the excitation–emission maps gradually become elongated along the diagonal, denoting the appearance of more red-shifted excitation–emission wavelengths. Moreover, a redshift of the excitation–emission maximum and an increase in the emission intensity is observed going from acidic to alkaline pH. We believe that such changes in the shape of the excitation–emission maps could be attributed to the reversible protonation and deprotonation of C-dot surface groups. In particular, under acidic conditions, C-dots can assemble into larger particles—as depicted by DLS measurements—and oxygen-containing moieties on their surface are oxidized, leading to a quenching of the photoluminescence of their surface emissive states. At basic pH values, the opposite trend is instead observed, leading to increased PL [26,45,46].

The pH-dependent map modification and the emission maximum shift and increase are more evident for C NH_4_OH and P NH_4_OH, while such features are less pronounced in the case of P H_2_O. Furthermore, there is no pH-dependent increase in the emission intensity of C H_2_O. This could suggest that the photoluminescence of C NH_4_OH and P NH_4_OH is mainly due to their surface emissive states, as pH determines the protonation/deprotonation of their surface groups. In contrast, for H_2_O samples, and in particular C H_2_O, the emission is less correlated with the surface properties and, therefore, more dependent on the core states of the C-dot.

When the QY will be improved, the pH-sensitivity of NH_4_OH samples would be of great interest, as pH-sensitive C-dots could be employed for intracellular pH sensing in place of poorly biocompatible luminescent nanomaterials such as, for instance, cadmium-based QDs [47]. Exploiting C-dots as pH sensors would, indeed, enable the monitoring of pH changes inside the cell and, thus, a deep insight into intracellular trafficking and functions [48]. Intracellular pH plays a fundamental role in maintaining normal cell activity, as it regulates several biological processes including cell proliferation, apoptosis, ion transport, endocytosis, and tumour growth. As reported in the literature, abnormal pH values can lead to a variety of diseases such as cancer and Alzheimer’s diseases [48]. For instance, normal cells maintain intracellular pH (pH_i_) at near-neutral values of ~7.2 and extracellular pH (pH_e)_ at ~7.4 [49]. However, cancer cells create a reversed pH gradient, with their pH_e_ being more acidic than that of normal cells. Cancer cells usually possess a higher pH_i_ of >7.4 and a lower pH_e_ of ~6.7–7.1 [50,51,52]. Thus, monitoring intracellular pH changes is of great importance not only for studying cellular functions but also to gain a better understanding of physiological and pathological processes.

In summary, we found that the chemical and the photophysical properties of our C-dots can be modulated depending on both the carbon source and the aqueous-based solvent used for the synthesis. Moreover, a marked pH-dependent luminescence was observed due to the protonation/deprotonation of surface groups, which may find application in intracellular pH sensing. To the best of our knowledge, no C-dots suitable for intracellular pH sensing produced via a simple carbonization process from food industry by-products have been reported up to now. Indeed, C-dots for intracellular pH sensing reported in the literature are usually obtained from chemical compounds such as citric acid and P-phenylenediamine, or by combining a carboxylic acid and an amine, including oxalic acid and *p*-phenylenediamine as well as acrylic acid and 1,2-ethanediamine [53,54].

### 3.3. In Vitro Biological Evaluation

C H_2_O was selected from all the C-dots samples for in vitro evaluation, as it has the highest quantum yield and can, therefore, be better applied for cell imaging.

C H_2_O was tested at the concentrations of 0.05, 0.125, 0.25, and 0.5 µg/µL in fibroblast cells using an MTT assay (Figure 7). Overall, no statistically significant differences in cell viability were observed during the analysis, confirming that C-dots do not induce cytotoxicity after 24 h of incubation in comparison to cells only.

Afterwards, the cellular uptake of C-dots was evaluated using microscope analysis, as illustrated in Figure 8. A fluorescence signal localized within cell cytoplasm can be observed, indicating that C-dots are being internalized into cells while preserving their luminescence. Overall, a dose-dependent increase in fluorescence and, thus, in cellular uptake is observed in comparison to cells only. The cellular uptake evaluation also confirmed an overall good cellular morphology, in agreement with cell viability evaluation. Therefore, fluorescence microscopy data prove the feasibility of C-dots from nut industry by-products as luminescent probes for cellular imaging.

## 4. Conclusions

We synthesized and characterized fluorescent C-dots via a carbonization process of P and C shells, thereby providing a new, sustainable approach to C-dot production in the framework of circular economy. As-synthesized C-dots possess excellent water dispersibility thanks to their negative surface groups, have good luminescence properties, and they have been successfully applied for in vitro cell imaging. Indeed, C-dots are not cytotoxic to fibroblasts and show good luminescence after cellular internalization. Moreover, our C-dots possess a marked pH-dependent luminescence due to the protonation/deprotonation of surface groups and may, therefore, be applied for intracellular pH sensing. We also proved that, by changing the source material and the dispersing medium, it is possible to alter the photophysical properties of C-dots. The application of the presented C-dots could be expanded to in vivo imaging as well as to the specific-targeting of cells and organelles. To this extent, future works should be focused on the synthesis of C-dots with improved quantum yield as well as on tuning their emission in the red region. Red-emitting C-dots have deeper tissue penetration, a lower level of interference from the autofluorescence of biomolecules, and negligible photodamage by the excitation light compared to blue-emitting ones. Furthermore, C-dots could be bio-functionalized using hyaluronic acid or hyaluronate to provide them with a cell-specific targeting moiety. Hyaluronic acid (HA) is widely used to functionalize nanoparticles and C-dots for the targeting of tumour cells through its interaction with cell-surface HA receptors (i.e., CD44 receptors) [55]. Indeed, cancer cells usually overexpress CD44 receptors and can, therefore, be targeted using HA-functionalized C-dots.

## Figures and Tables

**Figure 1 jfb-14-00090-f001:**
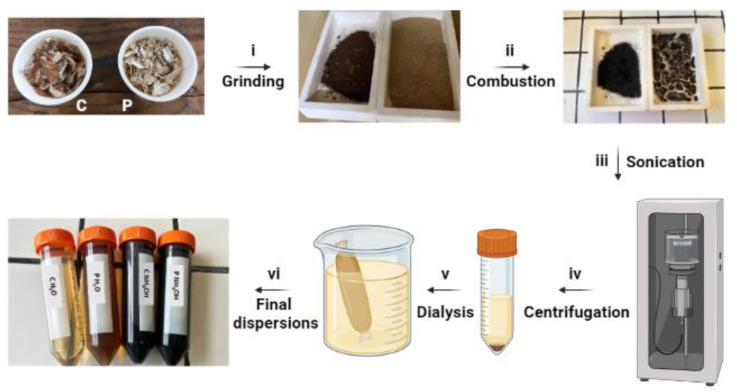
Synthetic process for C-dot production: (i) C and P shells were ground into fine powder, (ii) combustion, (iii) tip sonication, (iv) centrifugation, (v) dialysis, (vi) and final dispersions.

**Figure 2 jfb-14-00090-f002:**
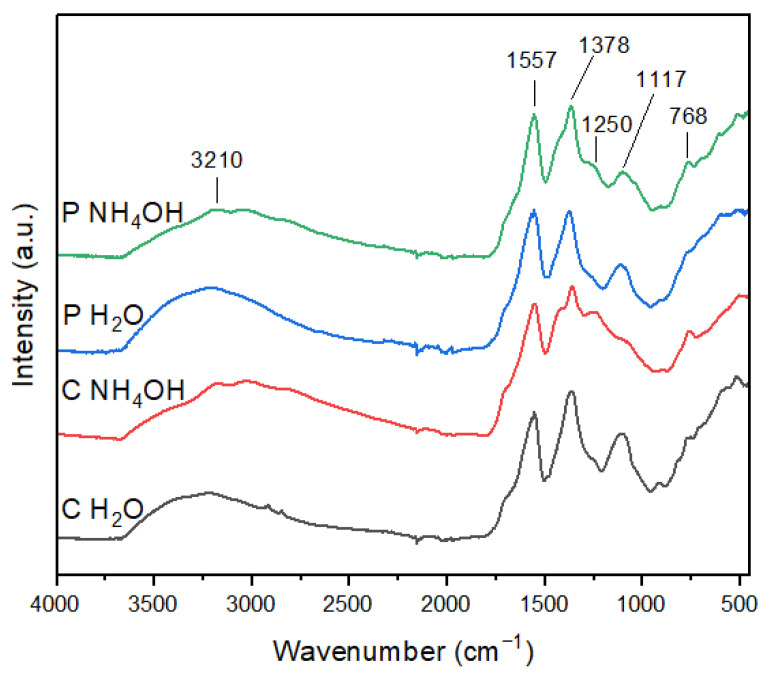
FTIR spectra of C-dots.

**Figure 3 jfb-14-00090-f003:**
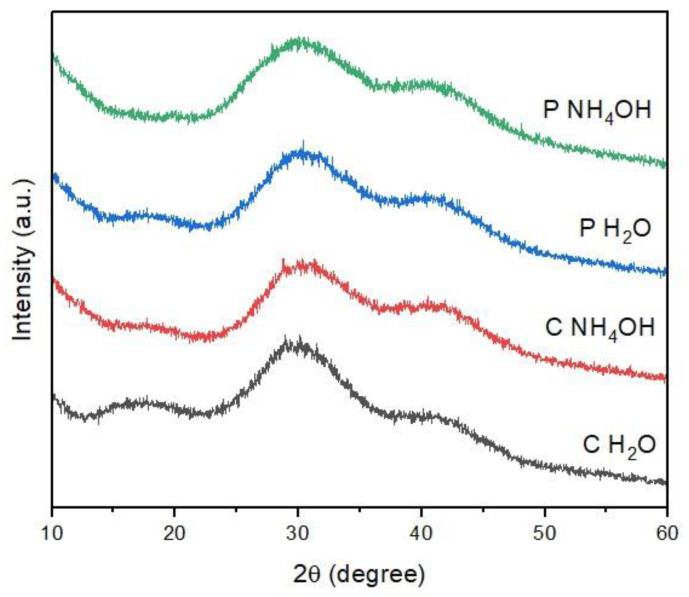
PXRD pattern of C-dots.

**Figure 4 jfb-14-00090-f004:**
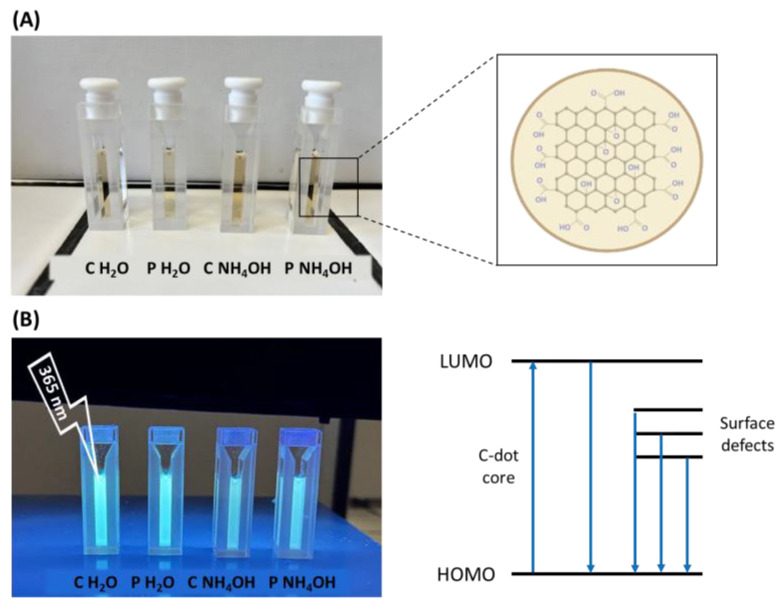
C-dots under daylight (**A**) and UV irradiation (**B**). Molecular orbitals and their related energy transitions are shown in Figure 4B as well.

**Figure 5 jfb-14-00090-f005:**
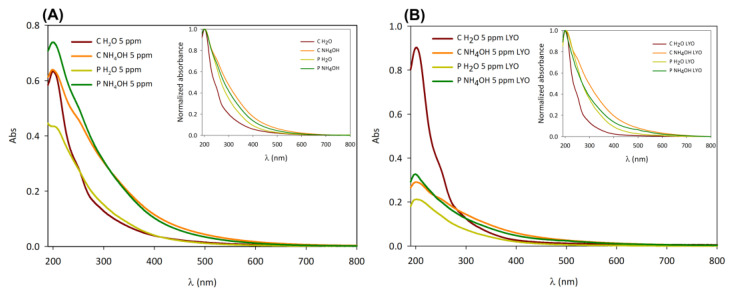
Absorption spectra of chestnut (C H_2_O and C NH_4_OH) and peanut (P H_2_O and P NH_4_OH) native dispersion (**A**) and freeze-dried (**B**) samples. Inset: normalized absorption spectra of native dispersion (**A**) and freeze-dried (**B**) samples.

**Figure 6 jfb-14-00090-f006:**
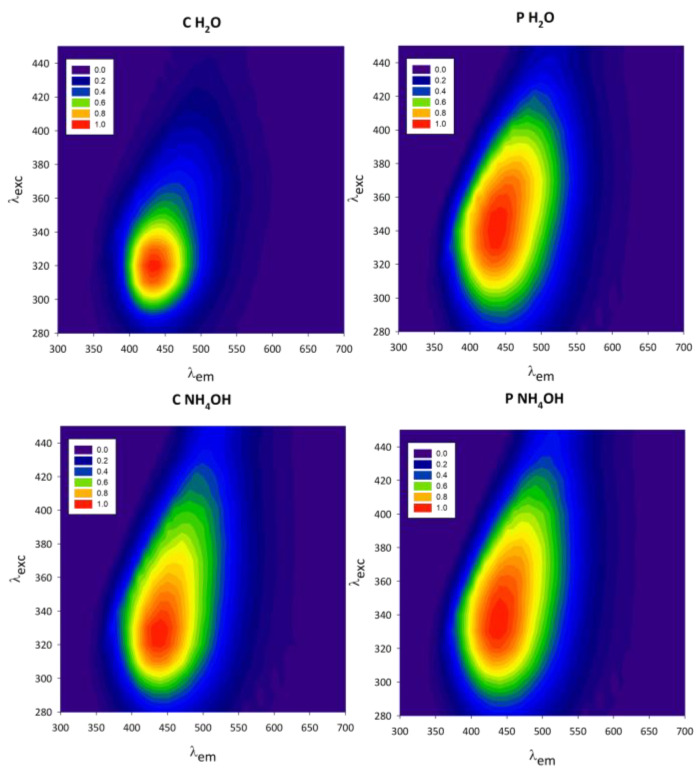
Normalized excitation–emission maps of chestnut (C H_2_O and C NH_4_OH) and peanut (P H_2_O and P NH_4_OH) native dispersions.

**Figure 7 jfb-14-00090-f007:**
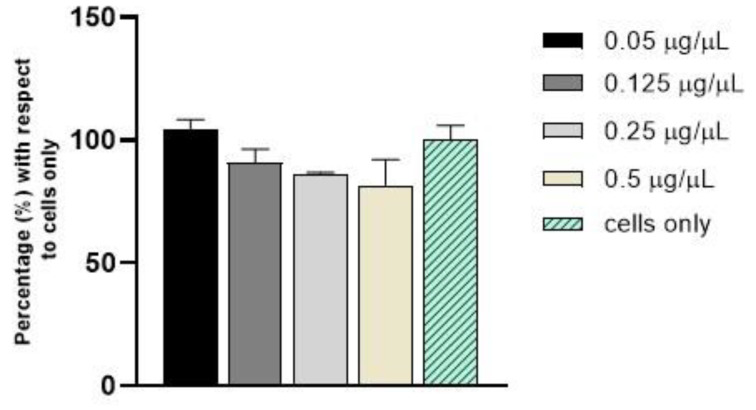
Cell viability analysis with MTT assay after 24 h of incubation with C-dots.

**Figure 8 jfb-14-00090-f008:**
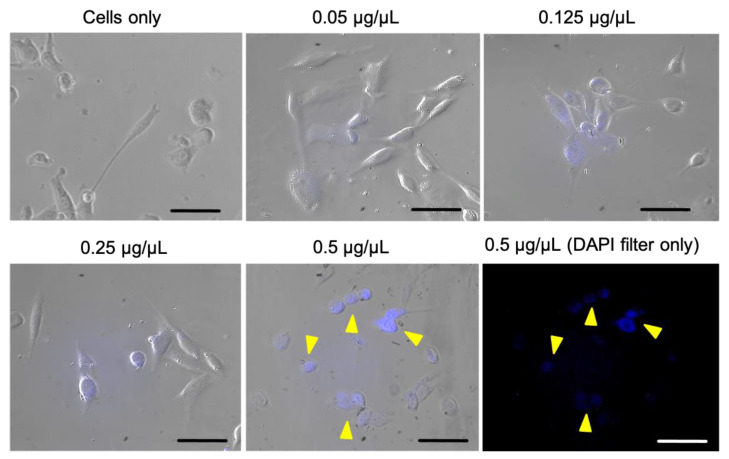
Cellular uptake after 6 h of incubation analysed by merging the bright field and DAPI filter (λem: 420 nm) images. In blue, C-dot internalization into cells (DAPI filter). Arrows indicate the C-dots in 0.5 µg/µL group. Scale bar is 50 µm.

**Table 1 jfb-14-00090-t001:** Hydrodynamic diameter and ζ-potential of C-dots.

Sample	Z-Average (nm) ± SD	PdI	ζ-Potential (mV) ± SD	Suspension pH
C H_2_O	106 ± 1	0.362	−26.7 ± 1	5.5
P H_2_O	102 ± 2	0.262	−21.2 ± 1	5.4
C NH_4_OH	114 ± 2	0.248	−51.7 ± 1	6.5
P NH_4_OH	165 ± 2	0.408	−22.6 ± 1	6.2

**Table 2 jfb-14-00090-t002:** Quantum yields and emission lifetimes of C-dots.

Sample	*Φ* (%)	τ (ns) at 450 nm	τ (ns) at 520 nm
C H_2_O	2.0	3.9	5.1
P H_2_O	1.4	3.7	4.5
C NH_4_OH	0.6	3.8	3.6
P NH_4_OH	0.9	4	4.4
C H_2_O lyo	4.0	4.5	5.2
P H_2_O lyo	1.6	3.8	4.3
C NH_4_OH lyo	0.6	3.6	4.4
P NH_4_OH lyo	0.9	4.2	4.6

## Data Availability

The data presented in this study are available in Appendix A.

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
