# Peer review of "Fluorescent Carbon Dots from Food Industry By-Products for Cell Imaging"

_jfb, 2023, doi:10.3390/jfb14020090_

Round 1
Reviewer 1 Report
Authors report the synthesis and characterization of emitting C-dots obtained from two different nut sources - chestnut and peanut shells through carbonization and extraction process. They investigated in detail the effect of two different carbon sources as well as of two aqueous-based solvents on the chemical and photophysical properties of C-dots. The main biomedical applications of the produced C-dots as well as future perspectives are discussed.
The work is certainly interesting, many results of characterization of the obtained C- dots have been obtained and can be published after some improvements:
1) There are a fairly large number of works on the synthesis and characterization of C-dots, including those obtained from waste, some of them are considered by the authors in the Introduction. Authors should point out the advantages of the C-dots obtained in the work in comparison with analogues.
2) The authors found that, depending on the type of nuts and the properties of the solution, there is a shift in λexc/λem. In this case, the luminescence quantum yield is 0.6-4.0%, which is lower than that observed in [22]. I did not find in the text at what concentration of C-dots the quantum yield was measured. I would like to draw the attention of the authors to the fact that the position of the emission maximum and the luminescence intensity (quantum yield) can depend on the concentration of C-dots in the solution (see, e.g. https://doi.org/10.1021/acsomega.1c01953). Authors should note it in article.
Author Response
Reviewer 1
Authors report the synthesis and characterization of emitting C-dots obtained from two different nut sources - chestnut and peanut shells through carbonization and extraction process. They investigated in detail the effect of two different carbon sources as well as of two aqueous-based solvents on the chemical and photophysical properties of C-dots. The main biomedical applications of the produced C-dots as well as future perspectives are discussed.
The work is certainly interesting, many results of characterization of the obtained C- dots have been obtained and can be published after some improvements:
1) There are a fairly large number of works on the synthesis and characterization of C-dots, including those obtained from waste, some of them are considered by the authors in the Introduction. Authors should point out the advantages of the C-dots obtained in the work in comparison with analogues.
We thank the Reviewer for the suggestion. The aim of our work was to contribute to the progress of C-dots preparation using carbon sources from circular economy. In particular, we used chestnut and peanut wastes as carbon sources and we evaluated in depth the chemical and photophysical properties of the resulting C-dots. In comparison to similar works we found out that the chemical and photophysical properties of our C-dots can be modulated on the basis of both the carbon source and the aqueous-based solvent used for the synthesis. Furthermore, we observed that the materials possess a marked pH-dependent luminescence due to the protonation/deprotonation of surface groups, which may find application in intracellular pH sensing. To the best of our knowledge there is no report on C-dots suitable for intracellular pH sensing produced by carbonization of food industry by-products. Indeed, C-dots for intracellular pH sensing reported in literature were obtained only from purified reagents, such as citric acid, P-phenylenediamine, oxalic acid, acrylic acid or 1,2-ethanediamine.
Therefore we have stressed out the advantages of our C-dots in comparison to the analogues reported in literature in the “Results and Discussion” session:
“In summary, we found out that the chemical and the photophysical properties of our C-dots can be modulated depending on both the carbon source and the aqueous-based solvent used for the synthesis. Moreover, a marked pH-dependent luminescence was observed due to the protonation/deprotonation of surface groups, which may find application in intracellular pH sensing. To the best of our knowledge, no C-dots suitable for intracellular pH sensing produced via a simple carbonization process from food industry by-products were reported up to now. Indeed, C-dots for intracellular pH sensing reported in literature are usually obtained from chemical compounds such as citric acid and P-phenylenediamine or by combining a carboxylic acid and an amine, including oxalic acid and p-phenylenediamine as well as acrylic acid and 1,2-ethanediamine[53], [54]”.
2) The authors found that, depending on the type of nuts and the properties of the solution, there is a shift in λexc/λem. In this case, the luminescence quantum yield is 0.6-4.0%, which is lower than that observed in [22]. I did not find in the text at what concentration of C-dots the quantum yield was measured. I would like to draw the attention of the authors to the fact that the position of the emission maximum and the luminescence intensity (quantum yield) can depend on the concentration of C-dots in the solution (see, e.g. https://doi.org/10.1021/acsomega.1c01953). Authors should note it in article.
We have now reported in the revised manuscript the concentration of C-dots used for acquiring the excitation/emission maps and for calculating the quantum yields. Both the excitation/emission maps and the quantum yields were obtained with the same concentration (i.e., 5 ppm) for all the tested C-dots. We reported it in the “Materials and Methods” section in paragraph 2.3 “C-dots characterization” and in the “Results and Discussion” section, in paragraph 3.2.3 “Quantum yield and emission lifetime”.
In the “Results and Discussion” section we have also commented about QY and maxima position changes in function of concentration as suggested by the Reviewer:
“Quantum yields were calculated at 450 nm to have a comparison of the emission properties of the different C-dots, and they are reported in Table 2. In particular, QY were obtained maintaining the same concentration for all samples (5 ppm), which is also the same as the one used for 3D excitation/emission maps: in this way, the possible concentration effects on their photophysical properties are avoided [44]”.

Reviewer 2 Report
In this manuscript, the authors reported the successful fabrication of carbon dots via thermal treatment of food industry by-products and demonstrated its potential as a luminescent probe for cellular imaging. Overall, the authors made a systemic investigation and try to use such carbon dots for biological application, and showed some positive outcome. The paper can be accepted for publication in Journal of Functional Biomaterials after a minor revision.
1. In pH-dependant Excitation-Emission maps, considering biological application scenarios, the authors should give some quantitative descriptions.
2. For cellular imaging, have the authors thought about bio-functionalization of carbon dots to improve the specific-targeting toward cells or organelles?
3. MTT assay seems to be not enough to address the biocompatibility of such carbon dots.
Author Response
Reviewer 2
In this manuscript, the authors reported the successful fabrication of carbon dots via thermal treatment of food industry by-products and demonstrated its potential as a luminescent probe for cellular imaging. Overall, the authors made a systemic investigation and try to use such carbon dots for biological application and showed some positive outcome. The paper can be accepted for publication in Journal of Functional Biomaterials after a minor revision.
- In pH-dependant Excitation-Emission maps, considering biological application scenarios, the authors should give some quantitative descriptions.
We thank the Reviewer for suggesting being more specific in this account. Indeed, pH plays a fundamental role in maintaining a normal cell activity, as it regulates crucial biological processes including cell proliferation, apoptosis, ion transport, endocytosis, and tumour growth. Abnormal pH values can lead to a variety of diseases such as cancer and Alzheimer’s disease (see, e.g. doi:10.3390/nano10040604) and, in turn, measuring pH can be a powerful tool to monitor these pathologies. For instance, Webb et al. found out cancer cells usually have a “reversed” pH gradient in comparison to healthy cells, which enables cancer progression (see, doi:10.1038/nrc3110). Thus, monitoring intracellular pH changes is of great importance not only for studying cellular functions but also to gain a better understanding of physiological and pathological processes.
We have highlighted these concepts in the “Results and Discussion” section:
“Intracellular pH plays indeed a fundamental role in maintaining a normal cell activity, as it regulates several biological processes including cell proliferation, apoptosis, ion transport, endocytosis, and tumour growth. As reported in literature, abnormal pH values can lead to a variety of diseases such as cancer and Alzheimer’s diseases[48]. For instance, normal cells maintain intracellular pH (pHi) to near-neutral values of ~7.2 and extracellular pH (pHe) of ~7.4[49]. However, cancer cells create a reversed pH gradient with their pHe being more acidic than that of normal cells. Cancer cells usually possess a higher pHi of >7.4 and a lower pHe of ~6.7–7.1[50], [51], [52]. Thus, monitoring intracellular pH changes is of great importance not only for studying cellular functions but also to gain a better understanding of physiological and pathological processes”.
- For cellular imaging, have the authors thought about bio-functionalization of carbon dots to improve the specific-targeting toward cells or organelles?
We thank the Reviewer for the suggestion. C-dots have recently attracted great attention in the field of cellular imaging owing to their excellent photophysical properties, low toxicity, and good biocompatibility. According to literature, the most reported bio-functionalization in order to impart a cell-specific targeting to C-dots involves the use of hyaluronic acid or hyaluronate (see, e.g. DOI: 10.1039/c6ra26048a). Hyaluronic acid (HA) is widely used to functionalize nanoparticles and C-dots for the targeting of tumour cells through its interaction with cell-surface HA receptors (i.e., CD44 receptors). Indeed, cancer cells usually overexpress CD44 receptors and can be therefore targeted using HA-functionalized C-dots.
We have envisioned this approach as future application of our C-dots, and discussed it in the “Conclusions” section:
“The application of the presented C-dots could be expanded to in vivo imaging as well as to the specific-targeting of cells and organelles. To this extent, future works should be focused on the synthesis of C-dots with improved quantum yield as well as on tuning their emission in the red region. Red-emitting C-dots have indeed deeper tissue penetration, lower level of interference from autofluorescence of biomolecules and negligible photodamage by the excitation light, compared to blue-emitting ones. Furthermore, C-dots could be bio-functionalized using hyaluronic acid or hyaluronate to provide them with a cell-specific targeting moiety. Hyaluronic acid (HA) is widely used to functionalize nanoparticles and C-dots for the targeting of tumour cells through its interaction with cell-surface HA receptors (i.e., CD44 receptors) [55]. Indeed, cancer cells usually overexpress CD44 receptors and can be therefore targeted using HA-functionalized C-dots”.
- MTT assay seems to be not enough to address the biocompatibility of such carbon dots.
The authors thank for the comment and agree with the Reviewer. In the present work only MTT assay has been used for a preliminary evaluation of the toxicity of C-dots. A more detailed characterization of cellular behaviour will be evaluated in the upcoming studies, once defined the best conditions for C-dots preparation. Therefore, we have corrected the title of the article, which now is “Fluorescent carbon dots from food-industry by-products for cell imaging”. Abstract, results and conclusions were also amended removing the word “biocompatible”.

Reviewer 3 Report
Carbon dots actually have great potential applications in biomedicine. In this manuscript, Mancini et al. claimed that they synthesized luminescent carbon dots with good biocompatibility via thermal treatment of chestnut and peanut shells for cell imaging. However, there have been many similar works, both in terms of preparation process and luminescence properties, this work lacks novelty. I cannot therefore recommend the adoption of the manuscript.
In addition, I don’t think the large-size carbon materials obtained by the authors can be called carbon dots. It is well known that the luminescence properties of carbon dots are closely related to their size and surface groups. I suggest the authors should first purify the products to obtain pure carbon dots, and then characterize them. Finally, additional in vivo imaging data are recommended.
Author Response
Reviewer 3
Carbon dots actually have great potential applications in biomedicine. In this manuscript, Mancini et al. claimed that they synthesized luminescent carbon dots with good biocompatibility via thermal treatment of chestnut and peanut shells for cell imaging. However, there have been many similar works, both in terms of preparation process and luminescence properties, this work lacks novelty. I cannot therefore recommend the adoption of the manuscript.
We apologize for not being clear enough on the novelty of our work. The aim of our investigation was to report an advance in C-dots preparation from natural sources in comparison to the current state of the art, both in terms of waste source material and photochemical characterization. While a few works on fluorescent C-dots from peanut shells waste are present in literature, the use of chestnut as carbon source represents an absolute novelty. In addition, in contrast to analogues reported in literature, we found out that the chemical and photophysical properties of our C-dots can be modulated by changing the carbon source or the aqueous-based solvent used during the synthesis. Furthermore, we also found out that our C-dots possess a marked pH-dependent luminescence due to the protonation/deprotonation of surface groups, which may find an interesting application in intracellular pH sensing.
Therefore, we have modified the “Results and Discussion” section to highlight the novelty of our work:
“Exploiting C-dots as pH sensors would indeed enable the monitoring of pH changes inside the cell and thus a deep insight into intracellular trafficking and functions[48]. Intracellular pH plays indeed a fundamental role in maintaining a normal cell activity, as it regulates several biological processes including cell proliferation, apoptosis, ion transport, endocytosis, and tumour growth. As reported in literature, abnormal pH values can lead to a variety of diseases such as cancer and Alzheimer’s diseases[48]. For instance, normal cells maintain intracellular pH (pHi) to near-neutral values of ~7.2 and extracellular pH (pHe) of ~7.4[49]. However, cancer cells create a reversed pH gradient with their pHe being more acidic than that of normal cells. Cancer cells usually possess a higher pHi of >7.4 and a lower pHe of ~6.7–7.1[50], [51], [52]. Thus, monitoring intracellular pH changes is of great importance not only for studying cellular functions but also to gain a better understanding of physiological and pathological processes.
In summary, we found out that the chemical and the photophysical properties of our C-dots can be modulated depending on both the carbon source and the aqueous-based solvent used for the synthesis. Moreover, a marked pH-dependent luminescence was observed due to the protonation/deprotonation of surface groups, which may find application in intracellular pH sensing. To the best of our knowledge, no C-dots suitable for intracellular pH sensing produced via a simple carbonization process from food industry by-products were reported up to now. Indeed, C-dots for intracellular pH sensing reported in literature are usually obtained from chemical compounds such as citric acid and P-phenylenediamine or by combining a carboxylic acid and an amine, including oxalic acid and p-phenylenediamine as well as acrylic acid and 1,2-ethanediamine[53], [54]”.
In addition, I don’t think the large-size carbon materials obtained by the authors can be called carbon dots. It is well known that the luminescence properties of carbon dots are closely related to their size and surface groups. I suggest the authors should first purify the products to obtain pure carbon dots, and then characterize them.
We did perform an extensive purification of C-dots, as reported in the Materials and Methods section: “the suspension was centrifuged (14000 rpm for 10 mins) and the supernatant containing C-dots was dialyzed against ultrapure water for 48 h (membrane MWCO = 3500 Da). After that, a further centrifugation step was performed (14000 rpm for 15 mins) and the resulting dispersion was filtrated over a 0.45 µm syringe filter”.
It must be considered that our aim was to study C-dots as-synthesized, without performing any post-synthesis surface functionalization. It is well reported that “naked” C-dots easily aggregate to form clusters in the aqueous-based solvent (see, doi: 10.1016/j.colsurfb.2015.04.012). For this reason, our C-dots are characterized by higher values of the hydrodynamic diameter than those usually reported in literature, where extensive functionalizations are often performed.
We have stressed out this point in the “Results and Discussion” section:
“The Z-average of the particles determined by DLS is reported in Table 1. The hydrodynamic diameter of C-dots ranges between 100 and 170 nm, with the two samples dispersed in NH4OH solution having a higher average diameter in comparison to the ones in water. Such values of the hydrodynamic diameter are higher than those usually reported in literature[9] and may be explained taking into account that we studied as-synthesized C-dots without performing any post-synthesis surface functionalization to improve their colloidal stability. Therefore, naked C-dots can easily aggregate to form clusters in the aqueous-based solvent, as previously reported in literature[26]. This tendency toward aggregation was also confirmed by TEM analysis, as it can be clearly observed in Figure S1a. Nevertheless, although infrequent smaller carbon nanoparticles between 10 and 20 nm in size can also be found, as evidenced by TEM and EDS (Figure S1b and S1c).
The high Z-average values of the four samples could also be explained considering the pH of the individual dispersions. Indeed, it has been reported in literature that pH values lower than 6, can induce C-dots aggregation into clusters and therefore be accounted for the high Z-average values as well[27]–[29]. Nevertheless, polydispersity (PdI) values are < 0.4 and similar for all C-dot suspensions, indicating a narrow particle size distribution for all materials”.
Finally, additional in vivo imaging data are recommended.
We thank the Reviewer for the advice, which we will definitely consider for future works. It must be taken into account that our paper represents a preliminary work on C-dots for medical imaging, and the obtained materials need to be optimized before performing any in vivo experiments.
We have mentioned this perspective in the “Conclusions” section:
“The application of the presented C-dots could be expanded to in vivo imaging as well as to the specific-targeting of cells and organelles. To this extent, future works should be focused on the synthesis of C-dots with improved quantum yield as well as on tuning their emission in the red region. Red-emitting C-dots have indeed deeper tissue penetration, lower level of interference from autofluorescence of biomolecules and negligible photodamage by the excitation light, compared to blue-emitting ones”.

Reviewer 4 Report
A well written paper about the synthesis of a more sustainable carbon dots production for biomedical applications of carbon dots of food industry by-products.
The experimental setup is well taken und the conclusions are comprehensible. The only point a like to see a small additional comment on figure 3. The PXRD pattern of the C-dots shows also different broad band at 17° 2θ. A comment about this behavior would be appreciated.
Author Response
Reviewer 4
A well written paper about the synthesis of a more sustainable carbon dots production for biomedical applications of carbon dots of food industry by-products.
The experimental setup is well taken und the conclusions are comprehensible. The only point I would like to see a small additional comment on figure 3. The PXRD pattern of the C-dots shows also different broad band at 17° 2θ. A comment about this behavior would be appreciated.
Following the Reviewer’s suggestion, we studied in more detail the band at 17° 2θ that has never been reported in the literature on C-dots. We found out that the band was due to PXRD instrument sample holder, as it disappeared after recollecting the pattern with a quartz sample holder (see new Figure 3). We have corrected Figure 3 with the new patterns collected for samples P H2O, C NH4OH and C H2O.

Round 2
Reviewer 3 Report
The manuscript has been revised mostly according to the comments and suggestions of the reviewers, and no further revision is needed. I recommend the acceptance of the manuscript for publication in JFB.